# Cryo-EM structure of single-layered nucleoprotein-RNA complex from Marburg virus

Luca Zinzula [1,2] ✉, Florian Beck[1,3], Marianna Camasta [1], Stefan Bohn [1,4], Chuan Liu[1], Dustin Morado [5,6], Andreas Bracher [7], Juergen M. Plitzko [1,3] & Wolfgang Baumeister [1,2] ✉

Marburg virus (MARV) causes lethal hemorrhagic fever in humans, posing a threat to global health. We determined by cryogenic electron microscopy (cryo-EM) the MARV helical ribonucleoprotein (RNP) complex structure in single-layered conformation, which differs from the previously reported structure of a double-layered helix. Our findings illuminate novel RNP inter-actions and expand knowledge on MARV genome packaging and nucleocapsid assembly, both processes representing attractive targets for the development of antiviral therapeutics against MARV disease.

Marburg virus (MARV) is an enveloped, non-segmented, single-stranded negative-sense RNA (ssRNA -) virus that belongs to the *Filoviridae* family, order *Mononegavirales*, and causes lethal disease in humans and non-human primates[1]. Its discovery dates to 1967, when in Marburg and Frankfurt (Germany), and in Belgrade (Yugoslavia, now Serbia), outbreaks of hemorrhagic fever originated as laboratory-acquired human infections from grivets (*Chlorocebus aethiops*) imported from Uganda[2,3]. Since then, MARV sporadically re-emerged throughout sub-Saharan Africa, where it circulates among Egyptian fruit bats (*Rousettus aegyptiacus*) serving as reservoir hosts[4]. Because of its international spread potential, MARV poses a threat to global health, and the concern is further exacerbated by the lack of approved vaccines and therapeutics against it[5,6]. Therefore, research aimed at investigating the MARV proteome to validate antiviral targets and to decipher the structural aspects of its mechanism of infection stands as the utmost priority. Among the seven proteins encoded by the MARV genome, the nucleoprotein (NP) exerts its fundamental role in encapsidating ssRNA into a helical ribonucleoprotein (RNP) complex, which in turn serves as scaffold for nucleocapsid formation, viral RNA transcription, and replication[7,8]. Recently, the MARV RNP complex structure from ectopically expressed NP truncation was determined by cryogenic electron microscopy (cryo-EM), revealing an arrangement of RNA-bound NP protomers into a double-layered, left-handed helix[9]. However, such double-helical conformation does not fully match with those previously shown by molecular architectures determined by cryo-electron tomography (cryo-ET), either from MARV-infected cells or intact MARV virions[10,11]. Moreover, since MARV RNP complexes were observed either as single helices or double ones in the cytoplasm of NP-expressing mammalian cells[8,9,12–14], questions remained on whether the double-helical structure would represent all aspects of MARV RNP complex formation. To fill this knowledge gap, in this work we report the detailed structure of MARV RNP complex in a more compact single-layer conformation, reconstituted in vitro by assembling recombinant NP onto synthetic ssRNA, imaged by cryo-EM and determined by single particle analysis (SPA) helical reconstruction.

## Results

### Reconstituted MARV RNP complex displays single-layer assembly

Following an experimental approach like the one that we previously adopted to determine the structure of the cetacean morbillivirus RNP complex[15], we fused a MARV NP truncation comprising its core domain (NP_core, residues 1–394) to a portion of its cognate chaperon, the virion protein 35 (VP35), comprising the NP-binding peptide (NPBP, residues

[1]Max Planck Institute of Biochemistry, Research Group Molecular Structural Biology, Martinsried, Germany. [2]iHuman Institute, ShanghaiTech University, Shanghai, China. [3]Max Planck Institute of Biochemistry, Research Group CryoEM Technology, Martinsried, Germany. [4]Institute of Structural Biology, Helmholtz Center Munich, Oberschleissheim, Germany. [5]Max Planck Institute of Biochemistry, Department of Cell and Virus Structure, Martinsried, Germany. [6]Stockholm University, Department of Biochemistry and Biophysics, Science for Life Laboratory, Stockholm, Sweden. [7]Max Planck Institute of Biochemistry, Department of Cellular Biochemistry, Martinsried, Germany. ✉e-mail: zinzula@biochem.mpg.de; baumeist@biochem.mpg.de

1–28)[16,17] (Fig. 1a). Via this strategy, full control over RNP complex formation can be achieved, since during protein expression the fused VP35 portion prevents NP from binding to cellular RNA, while post-purification removal of NPBP by a tobacco etch virus (TEV) protease renders NP capable of reconstituting an RNP complex in vitro upon incubation with synthetic ssRNA oligomers of six, or multiples of six, nucleotides (nt) in length (Fig. 1b). Purified uncleaved and NPBP-

cleaved MARV NP$_{core}$ proteins appeared highly homogeneous in sodium dodecyl-sulfate polyacrylamide gel electrophoresis (SDS-PAGE) analysis (Fig. 1c), and negative stain EM showed the former presenting as amorphous aggregates, whereas the latter oligomerized into ring-like structures of varying diameters. By contrast, when purified MARV NP$_{core}$ cleaved off from the VP35 NPBP was incubated with synthetic ssRNA corresponding to the first 18 nt of the MARV genome,

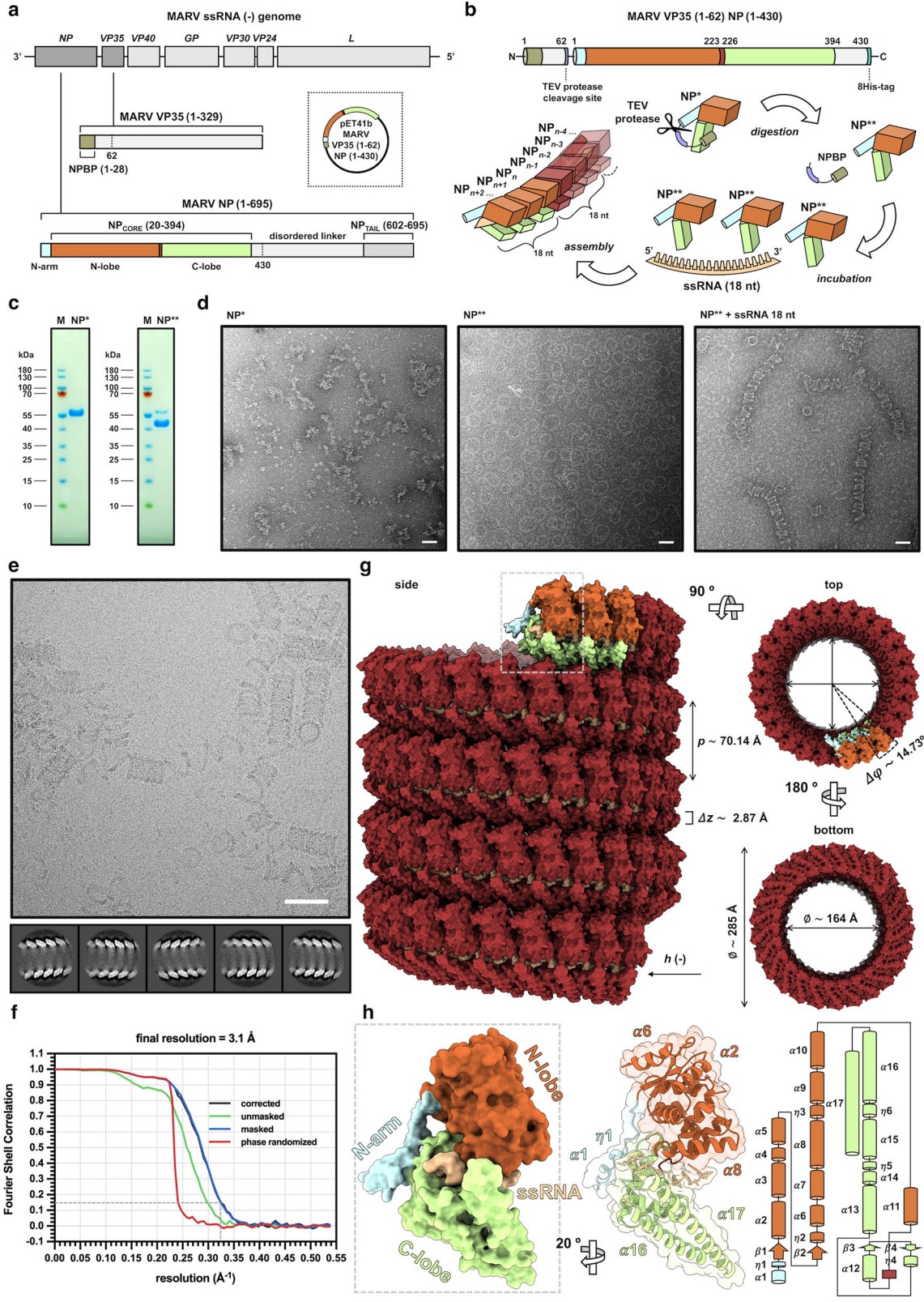

**Fig. 1 | Structure determination of MARV RNP complex in single-layer conformation. a** organization of MARV genome (upper panel) with *NP* and *VP35* genes highlighted in dark gray, and structural layout of encoded VP35$_{NPBP}$ and NP$_{core}$ (middle and lower panel, respectively) with subdomains highlighted in color (NPBP, avocado green; N-arm, light cyan; hinge, maroon; C-lobe, pastel green); dashed lines indicate boundaries of the recombinant construct used in this work, with a schematic representation of the corresponding expression plasmid vector shown in the inset. **b** structural layout (upper panel) of recombinant MARV VP35$_{NPBP}$ - NP$_{core}$ construct and a schematic representation (lower panel) of the workflow for the in vitro reconstitution of MARV RNP complex; single and double asterisks indicate fused and cleaved off VP35$_{NPBP}$, respectively; TEV cleavage and 8His-tag elements are highlighted in Melrose blue and downy-light-blue teal, respectively; subdomains of RNA-free and last three incorporated NP$_{core}$ protomers are colored as above, whereas previously assembled NP$_{core}$ protomers and ssRNA are highlighted in stiletto red and peach orange, respectively. **c** SDS-PAGE analysis of purified recombinant MARV VP35$_{NPBP}$ - NP$_{core}$, before (left panel) and after (right panel) digestion by TEV protease for VP35$_{NPBP}$ removal. **d** representative (*n* = 3) negative stain EM micrographs of recombinant MARV NP$_{core}$ fused to VP35$_{NPBP}$ (left panel) and devoid of it, featuring rings in the absence of (middle panel), or helices after incubation with (right panel) 18 nt ssRNA (scale bar, 50 nm). **e** representative (*n* = 5518) cryo-EM micrograph (upper panel) of in vitro reconstituted MARV RNP helical complexes (scale bar, 50 nm) and 2D classes of single particle averages (lower panel). **f** Fourier Shell Correlation (FSC) plot of the MARV RNP complex cryo-EM 3D reconstruction, showing a final resolution to 3.1 Å, as estimated by application of gold-standard 0.143 cutoff, indicated as a dashed line. **g** atomic model of MARV RNP complex determined from the cryo-EM density map, displayed as isosurface representation of three orthogonal views; values for the helical parameters (*h*, handedness; *p*, pitch; Δz, rise; Δφ, twist) and inner and outer diameter dimensions (⌀) are indicated; NP$_{core}$ subdomains of the last three incorporated NP$_{core}$ protomers, previously assembled NP$_{core}$ protomers and ssRNA are colored as above. **h** atomic model of a single ssRNA-bound NP$_{core}$ protomer of the MARV RNP complex, displayed as isosurface (left panel) and ribbon (middle panel) representations; a cartoon schematization of its secondary structure topology is shown (right panel); ssRNA and NP$_{core}$ subdomains are colored as above.

it reconstituted to the typical helical structures reminiscent of native MARV RNP complexes[8] (Fig. 1d). Similarly, in the vitreous ice of the plunge-frozen cryo-EM sample, the in vitro reconstituted MARV RNP complex formed cylinders made up of NP$_{core}$-ssRNA stacked spirals interspersed with tracts of relaxed coils resembling historic phone cords. Cylinders oriented perpendicularly to the optical axis were applied to SPA processing, and two-dimensional (2D) averaging and classification of their overlapping segments revealed a helical course with hollow interior and herringbone pattern (Fig. 1e and Supplementary Fig. 1). Interestingly, no double-layer arrangement was observed in any of the 2D-classes (Supplementary Fig. 2), and even re-classification of the particles from classes that were not displaying a clear polarity in their helical course led (apart from a few particles having too low signal-to-noise-ratio for assignment of the correct polarity) to class averages showing features of single-layer assembly only (Supplementary Fig. 3). Consistently, 3D reconstruction with helical symmetry from the best 2D classes, and refinement around an asymmetric unit composed by three NP$_{core}$ protomers bound to one 18 nt ssRNA molecule, produced density maps in single-layer conformation at 3.2 (helix) and 3.1 angstrom (Å) (asymmetric unit) final resolution (Fig. 1f and Supplementary Table. 1). Moreover, comparison between the 2D class averages and the real-space 2D projections of our density map with those of the map from the reported MARV RNP complex in double-layer conformation, deposited in the Electron Microscopy Data Bank (EMDB) as EMD-31420, and analysis of their power spectra, showed similar patterns of layer lines between classes and projections of our dataset, which are in turn remarkably different from the layer lines of the spectra from the double-layered MARV RNP complex projections (Supplementary Fig. 4). Furthermore, analysis of the helix diameter showed agreement, in their histogram diameter distribution, between the 2D classes and real-space density map projections of our dataset, which markedly differ from that of the double-layered MARV RNP complex, displaying a unimodal profile with distances between peaks that well match those measured for the inner and outer diameters in our reconstructed map (Supplementary Fig. 5).

### Structural features of single-layered MARV RNP complex

In agreement with the results from 2D classification and 3D reconstruction, atomic modeling of the MARV RNP complex structure describes a single-stranded, left-handed (*h* -) hollow helix with inner and outer diameters of 164 Å and 285 Å, respectively. Each helix turn is completed by ~ 25 NP subunits, and repeats with a pitch of about 70 Å (Fig. 1g). Moreover, with a uniformly represented angular distribution of the particle views that resulted in a local resolution spanning 3.0–3.5 Å, the cryo-EM density map had sufficient details to build an atomic model of the three ssRNA-bound MARV NP$_{core}$ protomers, with

fully distinguishable secondary structures comprising the N-terminal arm (N-arm, α1-η1), N-terminal lobe (N-lobe, β1- α11), linker (β4- η4) and C-terminal lobe (C-lobe, α12- α17) topologies, and a 18 nt-long segment of continuous ssRNA running along the groove between C-lobe and N-lobe subdomains for a tract of 6 nt per NP protomer (Fig. 1h and Supplementary Fig. 6). Superimposition of this structure with those of MARV NP$_{core}$ devoid of ssRNA or bound to VP35 NPBP[16,17] highlights the translation and the rotation undergone by the C-lobe during the shift from the RNA-free open state toward the RNA-bound closed one, which implies helix α17 (residues 351–390) elbowing inward against helix α16 (residues 322–349) and ssRNA, and helix η6 (residues 311–316) moving upwards beneath the nucleic acid, the latter being in turn protected at its upper side by helix α8 (residues 147–165) (Fig. 2a). Nevertheless, root mean square deviation (RMSD) values suggest that, apart from this conformational change, the overall structure remains unchanged beyond the local level compared to previously described MARV NP$_{core}$ models (Fig. 2b). In addition, a high level of residue conservation is observed among the NP$_{core}$ sequences from the various strains within the *Orthomarburgvirus* genus (Fig. 2c) and - especially for the regions spanning residues 132–147, 222–233 and 274–329 – also among those from other species of the *Filoviridae* family, consistent with the high structural similarity (Supplementary Fig. 7). Furthermore, one the most conserved NP$_{core}$ regions within marburgviruses encompasses residues 218–364, comprising a stretch of amino acids predicted to harbor a coiled-coil motif and reported as essential for viral RNA synthesis and RNP complex assembly[18]. Indeed, our structure revealed that an intra-chain, non-canonical parallel coiled-coil is formed in the MARV NP$_{core}$ between the helices α15 and α16, stabilizing the C-lobe in an orientation that favors both clamping of the ssRNA by helix η6 and NP$_{core}$ interaction with adjacent protomers via helix α16 (Supplementary Fig. 8). Noteworthy, although at the level of the single protomer our structure shares similar conformation with the double-layered RNP complex, at the level of biological assembly it markedly diverges, because of its single-layeredness, from the one in double-helix conformation previously described[9] (Fig. 3a), displaying different helical parameters and electrostatic surface potential distribution (Supplementary Fig. 9). In fact, while two types of vertically adjacent asymmetric subunits, namely NP-a and NP-b, alternately repeat every two turns in the double-layered MARV RNP complex[9], the single-layered helix is composed of NP subunits with a single conformation that vertically repeats at each turn. Thus, when inter-rung dimeric units from the two RNP complexes are superimposed, the one from the single-layered helix consisting of two consecutive NPs along the axial direction, and the one from the double-layered helix made up of two vertically adjacent NP-a and NP-b, the upper protomer of the former unit aligns with the NP-a of the latter one, whereas the lower protomer

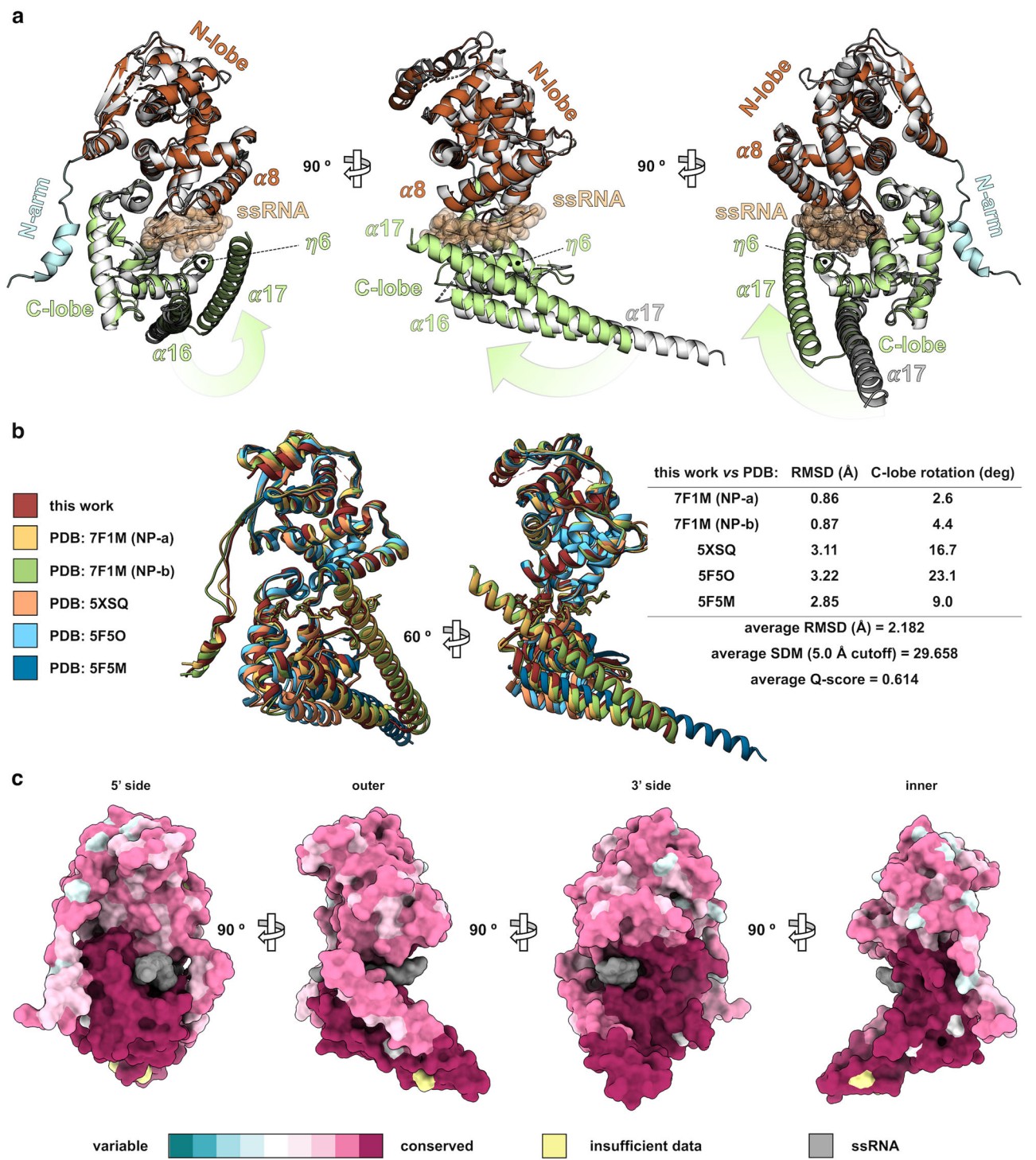

**Fig. 2 | Comparative structural analysis of MARV NP_core folding and evolutionary conservation. a** superposition of RNA-bound (this work, subdomains and ssRNA highlighted in color as in Fig. 1) with RNA-free MARV NP_core (PDB: 5F5M, light gray) displayed as ribbon representation in three orthogonal orientations; the NP_core conformational change consequent to ssRNA binding, involving N-lobe $\alpha$8 and C-lobe $\eta$6, $\alpha$16 and $\alpha$17, is highlighted by green arrows. **b** structural alignment of superposed all MARV NP_core structures available, including cryo-EM RNA-bound in single- (this work, stiletto red) and double-layer (PDB: 7F1M, golden tainoi-yellow-orange and wild-willow green for NP-a and NP-b, respectively) conformation, and X-ray crystallography RNA-free models (PDB: 5XSQ, atomic tangerine; PDB: 5F5O, Malibu blue; PDB: 5F5M, cerulean) displayed as ribbon representation in two

orientations (left panel); RMSD of matching C-$\alpha$ atoms, degrees of C-lobe rotation, structural distance measure (SDM) and secondary-structure matching (SSM) Q-score values for structural similarity evaluation are indicated in the table (right panel). **c** amino acid residue similarity calculated over all 143 sequences within the *Orthomarburgvirus* genus available in the Bacterial and Viral Bioinformatics Resource Center (BV-BRC) database (https://www.bv-brc.org, last accessed on May 8th, 2023) and mapped onto the MARV NP_core structure (this work) displayed as isosurface representation in four orthogonal orientations, 0 % to 100 % conservation is shown as blue-stone-teal to amaranth-deep-purple gradient scale, residue positions with insufficient data for conservation score calculation and ssRNA occupancies are shown in Canary yellow and dusty gray, respectively.

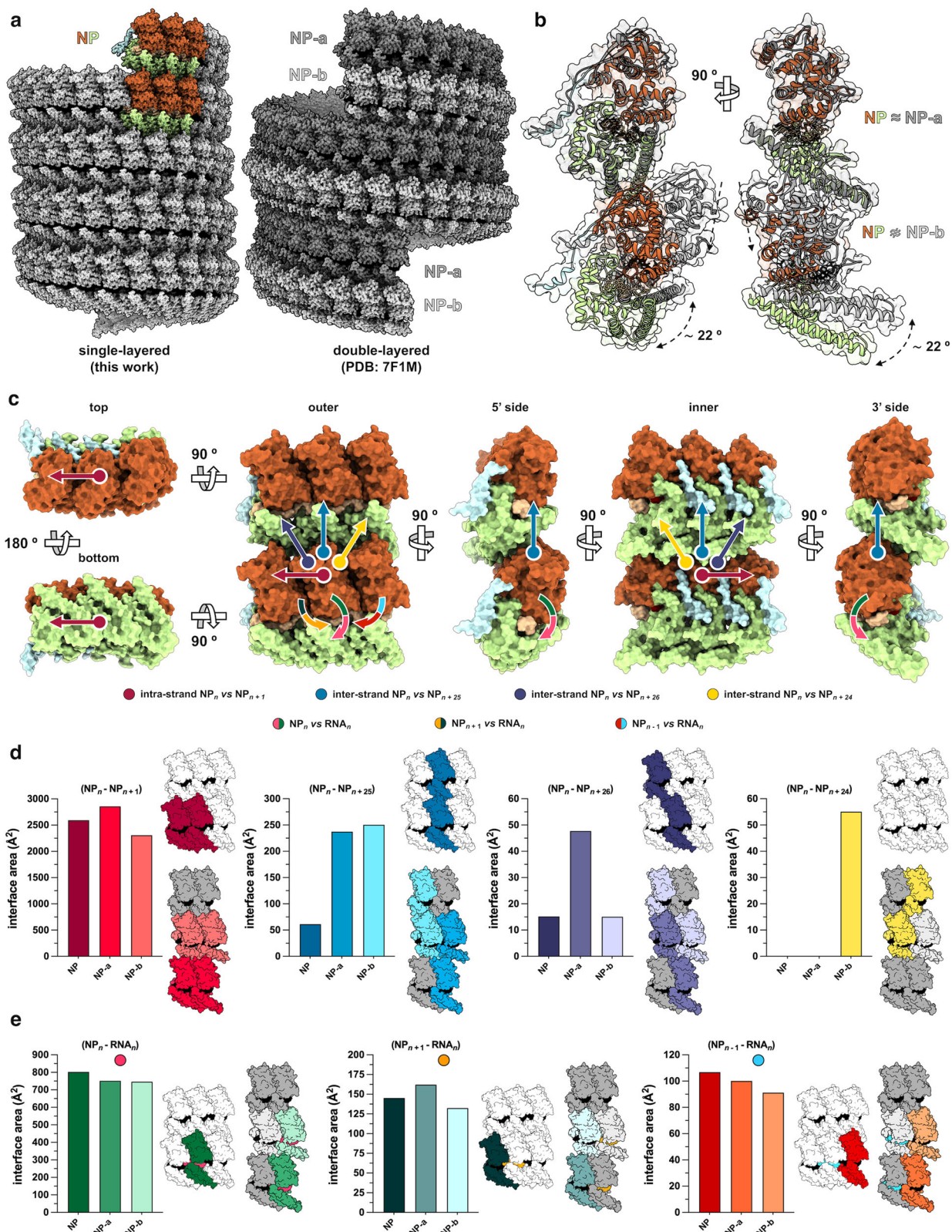

markedly misaligns from the NP-b due to a ~ 22° downward shift of the C-lobe and a lateral rotation of the N-lobe (Fig. 3b).

## NP-NP interactions in the single-layered MARV RNP complex

A hexameric minimal unit made up of adjacent NP$_{core}$ protomers from two consecutive rungs is sufficient to describe all protein-protein interactions in the helix, which are categorized as intra-strand (i.e.,

between any arbitrary NP$_n$ protomer and its neighbor NP$_{n+1}$), axial inter-strand (i.e., between any given NP$_n$ protomer and the one occupying the same position in the consecutive rung, i.e., NP$_{n+25}$) and non-axial inter-strand (i.e., between any given NP$_n$ protomer and those occupying the adjacent positions in the consecutive rung, i.e., NP$_{n+24}$ and NP$_{n+26}$) (Fig. 3c). Intra-strand interactions are of three types, the first one involving $\alpha 1$, $\eta 1$ helices and $\beta 1$ strand of NP$_n$ N-arm and N-lobe,

**Fig. 3 | Comparative analysis of interactions within MARV RNP complexes.**
**a** atomic model of cryo-EM structures of the MARV RNP complex in single- (left, this work) and double-layer (right, PDB: 7F1M) conformations displayed in side view as space filling-sphere representations; in the former model subdomains and ssRNA within an arbitrary minimal unit of three adjacent NP$_{core}$ protomers at the same positions of two consecutive rungs are highlighted with subdomains colored as in Fig. 1, whereas in the latter one NP-a and NP-b protomers of the two different strands are highlighted in shades of dark and light gray, respectively.
**b** superposition of minimal units consisting of axially adjacent NP$_{core}$ dimers, shown as ribbon representations, from MARV RNP complex in single- (protomers with colored ssRNA and subdomains) and double-layer (NP-a and NP-b protomers highlighted in dark and light gray, respectively) conformations; misalignment of the N-lobe and C-lobe subdomains in the lower strand is indicated by dashed lines and rotation angle. **c** atomic model of a hexameric minimal unit within the MARV

RNP complex in single-layer conformation (this work), consisting of three laterally and axially adjacent NP$_{core}$ protomers, displayed as isosurface representation in orthogonal orientations; functional subdomains and ssRNA are highlighted in colors as in Fig. 1; colored arrows define the different types of intra-strand and inter-strand interactions between NP$_{core}$ protomers and of NP$_{core}$ with each tract of 6 nt ssRNA. **d** comparative quantitative analysis of interaction areas between NP$_{core}$ protomers of MARV RNP complexes in single- and double-layer conformations, calculated for arbitrary hexameric minimal units of three and two adjacent NP$_{core}$ protomers, respectively, at the same positions in two consecutive rungs, covering all types of intra-strand and inter-strand NP$_{core}$-NP$_{core}$ interactions. **e** comparative quantitative analysis of interaction areas between NP$_{core}$ protomers and ssRNA of MARV RNP complexes in single- and double-layer conformations, calculated for arbitrary minimal units defined as above, covering all types of NP$_{core}$-RNA interactions.

$\alpha4$, $\alpha10$ and $\alpha11$ helices of NP$_{n+1}$ N-lobe, and $\alpha12$, $\alpha13$ helices and $\beta3$ strand of NP$_{n+1}$ C-lobe. The second one involves $\alpha5$, $\alpha9$, $\alpha10$ helices and $\beta2$ strand of NP$_n$ N-lobe, $\eta5$, $\alpha14$ and $\eta6$ helices NP$_n$ C-lobe, and $\alpha10$, $\alpha11$ helices of NP$_{n+1}$ N-lobe. The third one involves $\alpha13$ and $\alpha16$ helices of NP$_n$ C-lobe and $\alpha12$, $\alpha15$, $\alpha16$ and $\alpha17$ helices of NP$_{n+1}$ C-lobe. Inter-strand axial and non-axial interactions are established by the NP$_n$ N-lobe via $\alpha4$ helix and $\beta2$ strand, respectively, axially with $\alpha16$ helix of NP$_{n+25}$ C-lobe, and non-axially with $\alpha17$ helix of NP$_{n+26}$ C-lobe (Supplementary Fig. 10). Overall, intra-strand and inter-strand interactions that preside over the assembly of the single-layered MARV RNP complex rely on the same NP$_{core}$ regions as in the double-layered model. However, as a consequence of the single-layer arrangement, the structure described in this work differs by the number of residues and amino acid pairs involved in each type of interaction, some of which are unique to this conformation, at least within a 4 Å distance cutoff (Supplementary Figs. 11, 12). As a result, compared to those established between protomers in the NP-a and NP-b strands of the double-helical structure, contact areas in the single-layered MARV RNP complex are of intermediate value for the intra-strand NP$_n$ - NP$_{n+1}$ interaction, much lower and as low as for double-layer MARV NP-b value, respectively, for the inter-strand axial NP$_n$ - NP$_{n+25}$ and non-axial NP$_n$ - NP$_{n+26}$ ones, whereas no contacts - as in the case of double-layer MARV NP-a - within the hydrogen and van der Waals bonds 3-4 Å distance range give rise to any inter-strand non-axial NP$_n$ - NP$_{n+24}$ interaction (Fig. 3d).

### NP-RNA interactions in the single-layered MARV RNP complex
Interactions between NP$_{core}$ protomers and ssRNA are only intra-strand ones, established in such a way that any given hexameric RNA$_n$ is wrapped by the corresponding NP$_n$ protomer, but also interacts with the adjacent ones NP$_{n+1}$ and NP$_{n-1}$ (Fig. 3c). Overall, binding of each NP$_{core}$ protomer to a 6 nt-long tract of the 18 nt ssRNA in the asymmetric minimal unit of the single-layered MARV RNP complex follows the same modality, and involves the same set of highly conserved amino acid residues, as in the double-layered structure, with any hexameric RNA$_n$ tract being clamped by the NP$_n$ $\eta6$ and $\alpha17$ C-lobe helices to accommodate, twisted in a "three-bases-in, three-bases-out" conformation, into the groove beneath the NP$_n$ N-lobe. In this groove, ssRNA interacts, within a 4 Å distance range, with residues from the $\alpha8$ and $\alpha11$ helices of the NP$_n$ N-lobe, the $\alpha12$, $\alpha14$ and $\eta7$ helices of the NP$_n$ C-lobe, the $\alpha8$ and $\alpha11$ helices of the NP$_{n+1}$ N-lobe, and the $\eta6$ helix of the NP$_{n-1}$ C-lobe (Supplementary Fig. 13). Moreover, substantiating the rationale for a sequence-independent RNA-binding fashion, most interactions are electrostatics in character and established between the ssRNA phosphate backbone and basic amino acids such as Lys142, Lys153, Arg156 and Lys230, which were previously proven by muta-genesis as important - to various extent – for both RNP complex assembly and ssRNA synthesis[9]. Nevertheless, within the 4 Å distance cutoff, some interactions only appear in this assembly, involving the ssRNA phosphate backbone and Gln220 from the NP$_{n+1}$ N-lobe, and ssRNA bases and hydrophobic amino acids such as Pro141, Val144,

Val145, and Ala149 (Supplementary Figs. 13, 14). In addition, differences between the two RNP complex conformations exist in the extension of the ssRNA-binding surface, given that, compared to those between the nucleic acid and the protomers of the NP-a and NP-b strands in the double-layered RNP complex, the contact areas between NP$_{core}$ and ssRNA in the single-layered structure are much wider for the NP$_n$ - RNA$_n$ and NP$_{n-1}$ - RNA$_n$ interactions, whereas settle on an intermediate value for the NP$_{n+1}$ - RNA$_n$ one (Fig. 3e).

### Comparison with MARV virion and RNP complexes of other filoviruses
The atomic models and the cryo-EM density maps from our single-layered MARV RNP complex and the double-layered one reported by Fujita-Fujiharu and colleagues[9] can both be fit to the volume density map of a molecular architecture of the MARV nucleocapsid (EMD-3875) determined by cryo-ET from authentic virions[11], although with some differences. With regard to the size, the larger diameter and resulting smaller curvature make the double-layer conformation potentially more compatible and better fit to the nucleocapsid volume, with the caveat, however, that both RNP complex models are based on the NP$_{core}$, and thus such compatibility may not apply or be reversed in the context of a full-length NP. With regard to the helical pattern, in the case of the single-layered RNP complex, a plane defined between two rungs of the helix and coplanar with the nucleocapsid inter-rung groove passes through to cut off - about by half - the density of the outermost protrusions branching off from the protomers of the top rung. In contrast, in the case of the double-layered RNP complex, the same plane cuts through the nucleocapsid between the protrusions branching off from the rungs above and below it (Fig. 4). Noteworthy, such protrusions have been attributed to the nucleocapsid layer outside the RNP complex formed by VP24 and VP35[11,19], and recent work on Ebola virus (EBOV) has identified the outermost density of the protrusions as a third layer composed of NP protomers not bound to ssRNA and potentially involved in interactions with the VP40 matrix lattice[20]. Furthermore, being in a single-layer conformation, the in vitro reconstituted MARV RNP complex here described resembles the structural features of ortholog complexes from the closely related EBOV[21,22] and Lloviu (LLOV)[23] viruses. In fact, the four single-layered RNP complex structures show comparable values in the pitch and the number of subunits per turn, nevertheless vary in quaternary structure because of differences in the size of inner and outer diameters, as well as for the presence (EBOV and MARV) or the absence (LLOV) of non-axial inter-strand interactions between NP protomers (Supplementary Fig. 15).

### Discussion
In summary, our findings expand the current knowledge about MARV genome packaging and nucleocapsid assembly by providing an alternative structural framework of the RNP complex that may guide the development of structure-based antivirals against MARV disease. In

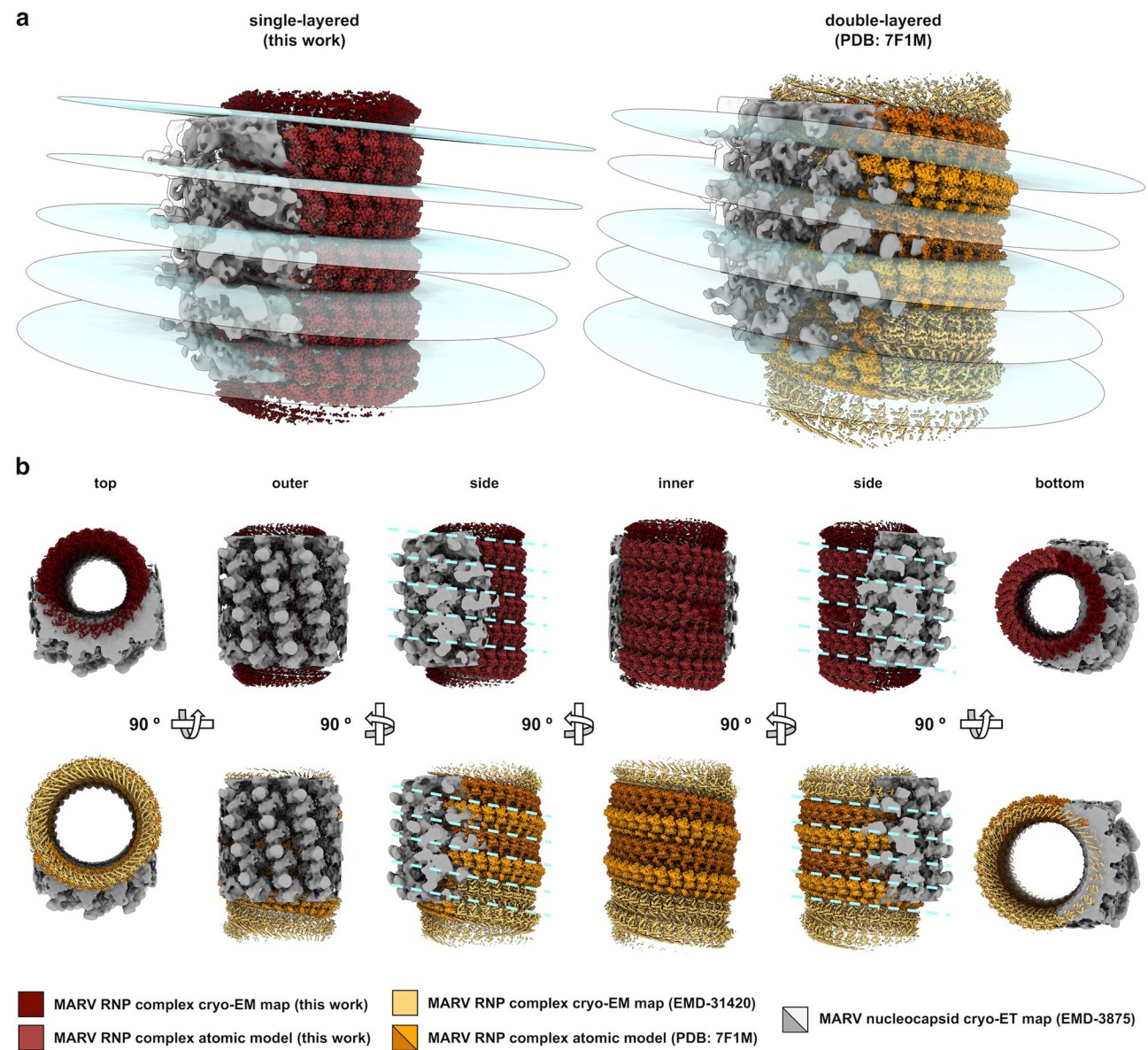

**Fig. 4 | Comparison between single- and double-layered MARV RNP complexes with nucleocapsid from MARV intact virion. a** Cryo-EM density map and atomic model of MARV RNP complexes in single- (left; this work, maroon, and stiletto red, contour map level 0.173) and double-layer (right; EMD-31420, PDB: 7F1M, golden-tainoi yellow-orange, peel-orange, tawny-orange, map contour level 4.7) conformations shown in side view, fitted into the cryo-ET density maps of MARV nucleocapsid from authentic virion (EMD-3875, dusty gray and transparent, map contour level 1.26 and 0.55, respectively) with planes across the helices, (glitchy-shader blue) defined to cut at the level of the groove between consecutive rungs, and highlighting coplanarity between rungs and differences in the helical course. **b** Fitting, as above, of the cryo-EM density map and atomic model of MARV RNP complexes in single- and double-layer conformation into the cryo-ET density map of MARV nucleocapsid (map contour level 0.55) shown as isosurface representation in six orthogonal orientations and colored as above; dashed lines (glitchy-shader blue) on the side views highlight the grooves that, in the respective models and the two maps, separate rungs at each helix turn.

this regard, subdomains responsible for the interactions between NP protomers and NP with ssRNA represent promising antiviral targets, as validated by the identification of chemical ligands capable of destabilizing or disrupting such interactions[24]. Moreover, the assembly of MARV RNP complex and its activity as a scaffold for viral transcription and replication could be targeted for high-throughput screening of drug candidates by coupling mini-genome systems with bioluminescence resonance energy transfer (BRET), or bimolecular fluorescence complementation (BiFC) assays, as it was recently reported for EBOV NP, and EBOV and MARV VP35 homo-oligomerization, respectively[25,26]. Furthermore, the single-layer conformation here described reconciles structural notions with respect to the apparent lack of consensus between the recently determined double-layered structure[9] and early

observations on MARV RNP complex morphology[8,10–13]. However, the fact that both single-layered RNP complexes and double-layered ones were observed after expression of MARV NP in mammalian cells[9] makes it plausible that metastable RNP assemblies with different packing and helical arrangement co-exist during MARV infection, raising interest in whether alternative conformations respond to different RNP complex functional states. Of note, a double-layer arrangement with two distinct asymmetric subunits NP-a and NP-b was also reported for an EBOV RNP-like complex apparently devoid of ssRNA[27], suggesting that such alternative assemblies may be common among filoviruses. In which circumstances and under which determinants single- and double-layered RNP complexes are formed during the filoviral life cycle remain to be elucidated and require further

investigation. Arguably, the MARV RNP complex in single-layer conformation is by itself compatible with both the roles as scaffold for replication and transcription in the cytoplasm and for genome packaging in the filoviral particle. Moreover, although in the latter case, it seems that a larger diameter would better fit into a mature virion, it is possible that during nucleocapsid assembly, the single-layered RNP complex undergoes adjustments in its helical parameters as a result of the interactions between NP and other nucleocapsid components. In addition, given the pleomorphic nature of MARV particles[8,10–13], it is plausible that one single-layered RNP complex may appear in different states of helix condensation, compactness, or relaxation along its entire length. In contrast, since a continuous double-layer conformation along the entire RNP complex implies the presence of two copies of the viral RNA genome that intertwine with each other into parallel left-handed helices, packaging of RNP complexes in such arrangement into mature nucleocapsids would result in genome diploidy of the budding virions. For EBOV, genome polyploidy was described in terms of multiple nucleocapsids per virion joined end-to-end in a modular fashion[28,29]. However, to the best of our knowledge, no structural evidence for the presence of two genomes within a single RNP complex has been reported thus far for MARV or any other filovirus. Alternatively, a fascinating hypothesis would be that filoviral double-layered RNP complexes may represent replicative intermediates that form locally in the cytoplasm during viral RNA synthesis, where two RNP complexes in single-layer conformations interact for a certain tract to form a double-layered tubular helix. Within this picture, it is tempting to anticipate that future in situ studies aiming at high-resolution information by cryo-ET will possibly help to elucidate the extent of structural diversity among RNP complexes from MARV and other filoviruses, in the context of either the cellular environment or the isolated virions.

## Methods

### Molecular cloning, protein expression, and purification

MARV (Mt. Elgon-Musoke strain) nucleotide reference sequences encoding for NP (YP_001531153.1) and VP35 (YP_001531154.1) were retrieved from the National Center for Biotechnology (NCBI) Protein database and used to obtain cDNA by synthetic preparation (BioCat, Germany). Domain boundaries were designed by bioinformatic analysis to place the VP35 region comprising NPBP (residues 1–62) upstream of the $NP_{core}$ (residues 1–430) with the insertion of a TEV protease recognition site (ENLYFQG) between the two domains for post-purification cleavage and VP35 NPBP removal, and the resulting chimeric construct was subcloned into a pET41b (Novagen) plasmid vector between the NdeI and XhoI restriction sites. Expression of the recombinant protein with a C-terminal hexahistidine (His₆)-tag was performed in E. coli BL21-DE3 (New England Bio-labs) grown in Terrific Broth medium (24 g L⁻¹ yeast extract; 12 g L⁻¹ peptone; 0.72 M $K_2HPO_4$, 0.17 M $KH_2PO_4$; 4 % v/v Glycerol) supplemented with 50 mg mL⁻¹ Kanamycin, at 37 °C and 200 rpm to an optical density at 600 nm of 0.8, then overnight (ON) at 24 °C by 0.65 mM isopropyl-β-D-1-thiogalactopyranoside induction. Harvested cells were lysed in buffer A (20 mM Tris-HCl, pH 8.5; 500 mM NaCl; 10 % v/v Glycerol; 1 mM DTT; 20 mM Imidazole) supplemented with 1 mg mL⁻¹ Lysozyme (Sigma-Aldrich), cOmplete EDTA-free Protease Inhibitor Cocktail (Roche), ~ 2000 Units Endonuclease from S. marcescens (Sigma-Aldrich), then sonicated and centrifuged for 30 min at 30,000 × g and 4 °C. The supernatant was subjected to affinity chromatography purification on Ni Sepharose High-Performance resin (GE Healthcare), with washing and elution steps performed with buffer A containing 40 mM and 800 mM imidazole, respectively. The eluate was dialyzed against Buffer A devoid of Imidazole, digested with TEV protease upon incubation for 30 min at 37 °C, and then subjected to size-exclusion chromatography (SEC) on a Superose 12 10/300 GL (GE Healthcare) column in buffer B (20 mM Tris-HCl, pH 8.0; 150 mM NaCl). Purity and homogeneity before and after TEV digestion were assessed by 4–12 % NuPAGE Sodium dodecyl sulfate–polyacrylamide gel electrophoresis (SDS-PAGE) (ThermoFisher), and concentrated to ~ 2.5 mg mL⁻¹ in Amicon Ultra centrifugal filter unit of 10,000 Da molecular weight cutoff (Merck-Millipore, Sigma-Aldrich).

### EM sample preparation

RNP complexes were reconstituted in vitro upon incubation of an excess of purified, TEV-digested MARV $NP_{core}$ with synthetic ssRNA oligomer (Metabion) resembling the first 18 bases of the MARV RNA genome (NCBI Nucleotide database reference sequence NC_001608; 5′ – AGACACACAAAAACAAGA – 3′) at ~ 3:1 protein-to-RNA molar ratio, for 24 h at room temperature (RT). For negative stain EM, MARV RNP complexes (~ 0.25 mg mL⁻¹), as well as TEV-digested and non-digested $NP_{core}$ negative controls (~ 0.025 mg mL⁻¹) were applied (~ 5 µL) to glow-discharged, carbon-coated 400 mesh nickel grids (Electron Microscopy Sciences) for 2 min at RT, then stained two times for 1 min and 30 s, respectively, with a 1:1 mixture of Methylamine Vanadate (Nano-Van, Nanoprobes) and Methylamine Tungstate (Nano-W, Nanoprobes), and air-dried. For cryo-EM, reconstituted ~ 4 µL reconstituted MARV RNP complex (~1.25 mg mL⁻¹) was applied to glow-discharged Quantifoil 1.2/1.3 grids, blotted for 3 s with force 1 at 95 % humidity and 4 °C, and plunge-frozen in liquid ethane cooled by liquid nitrogen in a Vitrobot Mark IV (Thermo Fisher).

### EM data collection

Negative stain data were acquired in a 200 kV Tecnai F20 (FEI) transmission electron microscope (TEM) equipped with a BM-Eagle 4 K CCD camera (FEI), operating with Serial EM software[30] at 62,000 × nominal magnification corresponding to 1.78 Å calibrated physical pixel size, with − 3.65 µm defocus and 1.2 s exposure, and single micrographs were saved for visual inspection with Fiji software[31]. Cryo-EM movies were acquired at 0° tilt angle in a 300 kV Titan Krios G4 Cryo-TEM G4 (Thermo Fisher) equipped with a Falcon 4 direct electron detector and Selectris X Imaging energy filter (Thermo Fisher), operating with EPU software (Thermo Fisher) at 130,000 × nominal magnification corresponding to 0.93 Å calibrated physical pixel size, applying ~ 40 e- Å⁻² total electron dose and − 0.5 µm to − 3.0 µm defocus range.

### Cryo-EM image processing, 3D reconstruction, and refinement

The alignment of 5518 movie frames and determination of the contrast transfer function (CTF) for the aligned frames were performed by RELION MotionCor implementation[32] and CTFFIND4[33], respectively. The helical reconstruction workflow of RELION 4.0[34] was applied to calculate the 3D structure of the RNP complex. For this, straight tubular sections were initially traced by hand from a random subset of 115 micrographs and used to train a crYOLO[35] network. The trained network served for the automatic picking of 67,385 particles from the entire micrograph dataset. Particles with a box size of 256 pixels (px) and a px size of 1.86 Å/px were extracted with helical priors and 16 asymmetrical units, which were used for the entire processing. The next steps consisted of two rounds of 2D classification into 200 and 100 classes, respectively, aimed at refining 2D averages and discarding particles that were poorly aligned or showed contaminations, which led to 26,416 particles in the first round and leaving 23,887 particles in the second one as final 2D dataset. To assess the level of heterogeneity in the dataset and rule out the co-existence of RNP complexes in single-layer and double-layer conformation, reclassification of the 2D classes that did not show a clear polarity in the helical course was performed on the remaining 23,887 particles from the last round of 2D classification. Averages that did not show a clear polarity were selected, accounting for 3956 particles, and subjected to another round of 2D classification into 200 classes, after which 747 particles (about 3 % of the final dataset from 2D classification) still did not show polarity in the resulting class averages. A starting model was generated by 3D

classification into three classes, using a cylindric reference and without imposing helical symmetry. The 3D class with the best resolution was selected, and helical parameters were determined using the RELION *relion_helix_toolbox*[34]. The twist was scanned from 4.0 to 18.0 degrees (deg) with an increment of 0.1 deg per search step, whereas the rise was scanned from 1.0 Å to 8.0 Å with an increment of 0.1 Å per search step, which led to the determination of a twist value of 16.0 deg and a rise value of 3.1 Å, respectively. Next, all particles were one class aligned with the selected class as reference and initial helical parameters from above, and the final step was a refinement with the starting model from one class refinement, which led to a 3.74 Å resolution. Hence, to improve the 3D classification, particles were CTF refined and Bayesian-polished after extraction to box 384 px and 1.24 Å px size, after which the consecutive refinement led to 3.4 Å resolution. Next, 3D classification into 10 classes without helical symmetry and with the starting model from refinement above was carried out. The two best-resolved classes were selected, and helical parameters were determined with *relion_helix*_toolbox[34] by twist scanning between 14.0 and 15.5 deg with 0.1 deg increments, and by rise scanning between 2.5 and 3.5 Å with 0.1 Å increments. This resulted in twist values of 14.1 deg and 15.3 deg and rise ones of 2.78 Å and 3.0 Å. Twist and rise of the selected classes were then averaged to 14.7 deg and 2.89 Å, respectively, and 17,118 particles were used for further processing. The next step was a refinement with the selected particles and the best resolved 3D class as a reference, which further improved the resolution to 3.3 Å. Helical parameters were refined with a twist scanning between 14.0 and 15.5 deg and increments of 0.05 deg, and a rise scanning between 2.7 and 3.1 Å with increments of 0.05 Å. The final processing steps for the entire structure consisted of CTF refinement, Bayesian polishing after extraction to box 512 px and 0.93 Å px size, and 3D refinement, which led to a resolution of 3.2 Å. Final helical parameters converged to a 14.73 deg twist ($\Delta\varphi$) and a 2.87 Å rise. Postprocessing of the map was performed by using DeepEMhancer[36] and RELION 4.0[34] software. To further improve the resolution, particles were symmetry expanded and subtracted using a mask around the NP$_{core}$ $n$ - 1, $n$, and $n$ + 1 protomers. This was followed by a new 3D refinement, which led to a 3.1 Å average final resolution, as well as a last post-processing step using DeepEMhancer[36] and RELION 4.0[34] software. Final maps were flipped to obtain the correct handedness, and local resolution estimation was performed using the built-in tool implemented in RELION 4.0[33] software.

## Helix diameter and power spectra analysis

For the comparative analysis of the power spectra and the inner and outer helix diameters between the maps of single-layered MARV RNP complex (this work) and double-layered one previously published by Fujita-Fujiharu et al. [9] and deposited in EMDB as EMD-31420, three stacks were created, which consisted of i) the 2D classes from the last classification round 1.86 Å px $^{-1}$, aligned with TOM-toolbox[37]; ii) the projections of the refined full 3D structure, scaled to 1.86 Å px $^{-1}$; and iii) the projections of the EMD-31420, also scaled to 1.86 Å px $^{-1}$. Projections were generated by using TOM-toolbox[37] with a tilt from 58 to 90 deg and rotation around the long axis from 0 to 180 deg with an increment of 8 deg, which led to 92 total projections. Power spectra were calculated using TOM-toolbox[37] and considering a $64 \times 64$ px cropped central portion, and logarithm was applied for display. For diameter analysis, 2D classes or projections were summed to generate individual intensity profiles, which were then combined and displayed. To measure the inner and outer helix diameters the first minima left and right to the global maximum were determined by using the *diff* function in MATLAB software. The outer diameter was calculated using the outer minima, whereas for the inner one the minima on the inner side of the global maximum was used.

## Model building and refinement

An AlphaFold2[38] atomic model of MARV NP residues 1–430 was placed into the map and manually edited with Coot[39]. RNA modeling started from the published model in Protein Data Bank (PDB, ID: 7F1M)[9]. To model the semi-continuous ssRNA density, nine adjacent copies of MARV NP$_{core}$ bound to three ssRNA 18-mer were first refined in real space with Phenix[40]. From the refined central chains of MARV NP$_{core}$ and the ssRNA 18-mer, a synthetic model was constructed, modeling the exact helical symmetry and the ssRNA longitudinal disorder (i.e., with + 6 nt and +12 nt register shifts). Residues with disordered side-chain were truncated at the C-beta. Residues at positions 105–107 and 120–126 of MARV NP were not included in the model because of missing density. The synthetic model was used to construct the helical filament with Lsqkab[41].

## Structural bioinformatics analysis

Molecular graphics were produced with PyMOL (http://www.pymol.org/), Chimera[42], ChimeraX[43], and GraphPad Prism (https://www.graphpad.com/scientific-software/prism/) software. For structural alignments, superposition of atomic models, and fitting of cryo-EM density maps, the TM-align[44], MatchMaker Match-Align[45], and Chimera[42] fit-in-map tools were used, respectively. For the calculation of Poisson−Boltzmann electrostatic surface potential, the PyMOL-embedded PARSE force field in PDB2PQR[46] and APBS[47] software tools were used. Mapping of the NP$_{core}$ secondary structure topology on orthologs within the family *Filoviridae* was performed by using the ESPript[48] server, run using as input for multiple sequence alignment (MSA) with the Clustal Omega[49] server the NCBI Protein database references YP_001531153.1 (*Othomarburgvirus marburgense*, Marburg virus, MARV), YP_009055222.1 (*Othomarburgvirus marburgense*, Ravn virus, RAVV), NP_066243.1 (*Orthoebolavirus zairense*, Ebola virus, EBOV), YP_138520.1 (*Orthoebolavirus sudanense*, Sudan virus, SUDV), NP_690580.1 (*Orthoebolavirus restonense*, Reston virus, RESTV), YP_003815423.1 (*Orthoebolavirus taiense*, Taï Forest virus, TAFV), YP_003815432.1 (*Orthoebolavirus bundibugyoense*, Bundibugyo virus, BDBV), YP_009513274.1 (*Orthoebolavirus bombaliense*, Bombali virus, BOMV), YP_004928135.1 (*Cuevavirus lloviuense*, Lloviu virus, LLOV) and YP_010087183.1 (*Dianlovirus menglaense*, Mênglà virus, MLAV). Mapping of residue conservation was performed by the ConSurf[50] server, considering all available NP amino acid sequences within the *Orthomarburgvirus* genus in the NCBI Bacterial and Viral Bioinformatics Resource Center (BV-BRC)[51] database (https://bv-brc.org; 143 returns, as of 8 May 2023). Coiled-coil analysis of conformation, register, knobs-into-holes packing, inter-helix, and knobs angles was performed using the SOCKET2[52] server, by setting parameters to 8.0 Å and 2.0 value the packing cutoff and helix extension parameters, respectively. Comparative mapping of residue contacts and quantitative analysis of interaction surface was performed by using COCOMAPS[53] server, setting to 4.0 Å the cutoff for interaction distance and using as input hexameric minimal units of single-layered (this work, three adjacent NP$_{core}$ protomers, same positions in two consecutive turns) and double-layered (PDB: 7F1M, two adjacent NP$_{core}$ protomers, same position in three consecutive turns)[9] MARV RNP complexes. For comparative analysis of the helical parameters among filoviral RNP complexes, structures of EBOV (PDB: 5Z9W; 6C54; 6NUT)[19,20,27] and LLOV (PDB: 7YPW; 7YR8)[21] RNP complexes were retrieved from the PDB and visualized with ChimeraX[43] software. For superposition of the MARV single-layered (this work) and double-layered (PDB: 7F1M, EMD-31420)[9] RNP complex atomic models and density maps with the MARV nucleocapsid (EMD-3875)[11] density map, the ChimeraX[43] built-in tool fit-in-map was used; cuts between rungs across the fitted models and maps were defined by planes let passing through the carboxyl oxygen (OD2) of Asp95 as the uppermost residue of the NP$_{core}$ protomer, for three adjacent protomers taken in the same relative positions at each

helix turn, and drawn by using the built-in tool axes/planes/centroids in ChimeraX[43] software.

## Reporting summary
Further information on research design is available in the Nature Portfolio Reporting Summary linked to this article.

## Data availability
The atomic coordinates of the single-layered MARV RNP complex structure generated in this work have been deposited in the PDB with the accession code 9FVD; the corresponding cryo-EM density maps have been deposited in the EMDB with the accession codes EMD-50803 and EMD-50804, for the helical assembly and the RNA-bound trimeric unit, respectively. For comparative analysis, atomic coordinates from other works, including those with accession codes 5F5M, 5F5O, 5XSQ, 5Z9W, 6C54, 6NUT, 7F1M, 7YPW and 7YR8, and cryo-EM and cryo-ET density maps including EMD-31420 and EMD-3875, were obtained from PDB and EMDB, respectively. Any other raw data related to this work are available from the corresponding authors upon reasonable request. Source data are provided in this paper.

## Code availability
A GitHub repository containing the code used in the 2D analysis of helix diameter and power spectra is available at https://github.com/FlorianBeckOle/2dClassAnalysis.git.

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

## Acknowledgements

Final cryo-EM dataset used to determine the structure herein presented was acquired using the infrastructure of the Department of Cell and Virus Structure at the Max Planck Institute of Biochemistry (MPIB) in Martinsried, Germany. We thank our colleagues at the MPIB, and we are particularly grateful to I. Nagy, D. Bollschweiler, D. Hrebik and J. Briggs for fruitful discussion and their excellent support.

## Author contributions

L.Z. conceived the study, designed the construct, and performed molecular cloning. L.Z., M.C., and C.L. performed protein expression and purification, carried out biochemical experiments, performed EM specimen preparation, and acquired negative stain EM data. L.Z., S.B., and D.M. performed cryo-EM dataset acquisition. L.Z. and F.B. performed cryo-EM image processing. A.B. performed structure atomic modeling. L.Z., F.B., and A.B. performed EM structural analysis and data interpretation. J.M.P. supervised EM workflow and data acquisition. W.B. supervised experimental design and data interpretation. L.Z. drafted the manuscript. All authors contributed to experimental design, data analyses, and manuscript review.

## Funding

## Competing interests

The authors declare that they have no competing interests.
