## [Transparent Peer Review file · Nature Communications]

Cryo-EM structure of single-layered nucleoprotein-RNA complex from Marburg virus.

Corresponding Author: Dr Luca Zinzula

Version 0:

Reviewer comments:

Reviewer #1

(Remarks to the Author)

Reviewer Remark to author :

The manuscript entitled "Cryo-EM structure of single-layered nucleoprotein-RNA complex from Marburg virus" from Luca Zinzula et al reported the structure of MARV RNP complex in a compact single layer conformation. The complex was recreated in vitro by assembling recombinant NP to synthetic ssRNA. It's intriguing that while previous research had shown a MARV double-layered left-handed helix, however the cyro-ET molecular structures match the single-layered conformation other than double-layered conformation. Luca Zinzula et al also discovered that the RNP assembly and ssRNA associative modality depend on the same regions as in the double-layered model. This makes a significant contribution to viral RNA transcription and replication as well as RNP creation. Though this is a nice work, a few suggestions are listed below to improve the current version.

1. Although I believe your work to be thorough and full, I advise making the discussion part stronger. Do the two RNP conformations that are demonstrated in vitro, for instance, actually exist in the natural state? If so, what physiological relevance does that have? If not, what external forces result in its formation?
2. In the course of your research, did you not notice the double-layered model at all during your research? Could it be that the proportion is too little to be taken into account during classification?
3. What distinguishes your findings from those of the Ebola virus RNP, which likewise has a single-layer spiral structure?
4. The addition of short ssRNA can cause NP to take on this helix structure, I wonder have you experimented with different nucleic acid lengths, and what were the outcomes?
5. Please clarify whether it is possible to design tests to demonstrate the natural activity of RNP.

Reviewer #3

(Remarks to the Author)

The manuscript by Zinzula et al. presents a cryo-EM structure of the Marburg virus nucleocapsid core in a single-helical state. The determined structure is particularly interesting because it strongly contrasts with the double-helical assembly described last year by Fujita-Fujiharu et al. Furthermore, the presented helical arrangement is common for nucleocapsids of non-segmented negative strand RNA viruses and closely resembles in particular the one of another famous filovirus, EBOV. If the EMD-31420 map had not been published, the structure presented in the manuscript of Zinzula et al. might be sufficiently convincing. Admittedly, one can always suggest that the glaring differences between the EMD-31420 and the presented EMD-17838 maps lie in the method of sample preparation. However, even though Fujita-Fujiharu et al. write that «the double-helical structure reconstituted in this study may not faithfully recapitulate the structure in the MARV

nucleocapsid core», given the current controversy surrounding the structure of the MARV nucleocapsid core, the authors of the present manuscript should provide more comprehensive evidence to support their conclusions. Can it be that both single-helical and double-helical structures coexist in their preparation, as it was seemingly the case for Fujita-Fujiharu et al.?

For instance, considering that the data was acquired on a Titan Krios G4 with a Falcon 4, it would have been appropriate to collect more micrographs (say 20,000-30,000 instead of 5,518) in order to perform a thorough 2D classification, beyond the one shown in Supplementary Figure 1. Indeed, one clearly sees from the Figure 1 that the preparation is extremely heterogeneous (which is quite common for the nucleocapsids of non-segmented negative strand RNA viruses), with very few helical filaments per micrograph suitable for high resolution image analysis. Importantly, in such cases it is crucial to specify how exactly the helical segments were selected, i.e. what was the box size and the percentage of overlap between consecutive segments. This information is missing in the manuscript, and in the figures sometimes a segment of one single turn is shown, sometimes 3, 4, 5 or 6. What was the segment size and overlap used for analysis, and were they changed during the processing or kept constant?

Following the first round of 2D classification shown in Supplementary Figure 1, one could select particles corresponding to classes showing helical side views and reclassify them iteratively, to see if all particles have the same diameter or if there is a heterogeneity at this level. To this end, it is essential to provide and analyse the histogram of diameter distribution (is it unimodal, centered on the inner-outer diameter of 174/299 Å of the presented EMD-17838 map, or bimodal, with a certain percentage of detected diameters corresponding to 255/330 Å as in EMD-31420, or are there also intermediate values present?). How do the sums of power spectra of particles in each class look like, which layerlines do they show and how different or similar they are between classes? Which differences are expected between a single-helical and a double-helical structures?

Idem, the 3D classification and the generation of the initial model is insufficiently documented. The starting model was generated by 3D classification imposing 3 classes and a cylindrical reference, with the helical parameters of Fujita-Fujiharu et al. as an initial guess. The authors write in materials and methods that the the class with the best resolution was selected as starting reference. Is this the class shown at the right side of the Supplementary Figure 1, under the 2D class averages? How many particles are present in this class? What do the other two classes look like? The authors then performed three rounds of 3D classification, 3D refinement, Bayesian polishing, and CTF-refinement, pooling the classes with the best resolution after each round, and resulting in a final dataset of 12,135 particles. Since no information on the shapes of the excluded 3D classes is provided, the presence of a low amount of particles corresponding to a double-helical structure cannot be excluded.

Since the structure solved correspond to NPcore-RNA only, please refrain from using the term nucleocapsid to avoid confusion with the full-length NP-RNA.

Line 48: «...ssRNA oligomers of six, or multiples of six, base pairs (bp) in length». You are using single strand RNA (ssRNA) with a length of six or multiple of six bases and not base pairs (there is no base pairing because it is a single strand RNA). Please replace «bp» by «bases» or «nucleotides» throughout the manuscript.

It is inappropriate to say that «the in vitro reconstituted MARV RNP complex formed cylinders made up of NP-RNA stacked rings» while meaning a helical filament. Of note, some paramyxoviral and pneumoviral nucleocapsids can indeed form both helices and stacked rings, therefore employing inaccurate terminology may contribute to further confusion.

At each helix turn, approximately 25.4 NP protomers follow one another with 14.1482° twist and 2.75 Å rise, leading to a spiral with 69.85 Å pitch». This sentence is unnecessary convoluted. In helices with one protomer per asymmetric unit, as the authors consider here because they don't take the RNA into account, there is no need to specify that the protomers follow one another. Instead, simply providing the values of twist and rise (or number of subunits per turn and pitch) is adequate.

The authors assembled NPcore on 18 base-long single strand RNA, with one NPcore bound to 6 consecutive bases. Therefore, in principle, if one considers the NPcore-RNA unit as an asymmetric unit of the helix, each asymmetric unit is composed of three NPcore protomers bound to one RNA segment. If the image analysis is performed with an asymmetric unit of 3 NPcore protomers, it would provide information about potential specific interactions with the RNA bases and the position of the base number 1, as previously done for paramyxoviruses (where an RNA of 6 bases was used for assembly and therefore the asymmetric unit was composed by only one N-RNA, thereby slightly simplifying the analysis). Hence, in order to fully exploit the potentials of their sample preparation strategy and gain novel information about the RNA binding beyond the one currently inferred from averaging over all protomers, it is highly advisable to revisit the analysis with three NPcores per asymmetric unit. This would imply modifying the RNA model building and refinement accordingly.

The authors claim that their reconstructed «...volume fits very well in that of the electron density map of a nucleocapsid molecular architecture determined by cryo-ET from authentic MARV virions (Supplementary Fig. 9)». First, to use the correct terminology, this is not an electron density map but a Coulomb potential map (or simply a cryo-ET map to avoid complicated terms). Second, this figure is not very convincing and the comparison between the maps deserves more explanation in the figure legend and the materials and methods section.

Finally, in the field of non-segmented negative strand RNA viruses, the N- and C-terminal domains of the nucleoprotein are called either N- and C-terminal domains, or N- and C-terminal lobes, or NTD and CTD. Please don't introduce new terms like foot and head, especially since from a structural point of view they cannot be applied consistently to the entire family.

Version 1:

Reviewer comments:

Reviewer #1

(Remarks to the Author)

I want to thank the authors for addressing my initial comments. Following the revision to the article, I do not have more

questions now.

Reviewer #4

(Remarks to the Author)

Manuscript Number: NCOMMS-23-36158-T- Cryo-EM structure of single-layered nucleoprotein-RNA complex from Marburg virus.

Point-by-point reply to the reviewers:

Reviewer #1

(Remarks to the Author):

Reviewer Remark to author:

The manuscript entitled "Cryo-EM structure of single-layered nucleoprotein-RNA complex from Marburg virus" from Luca Zinzula et al reported the structure of MARV RNP complex in a compact single layer conformation. The complex was recreated in vitro by assembling recombinant NP to synthetic ssRNA. It's intriguing that while previous research had shown a MARV double-layered left-handed helix, however the cryo-ET molecular structures match the single-layered conformation other than double-layered conformation. Luca Zinzula et al also discovered that the RNP assembly and ssRNA associative modality depend on the same regions as in the double-layered model. This makes a significant contribution to viral RNA transcription and replication as well as RNP creation. Though this is a nice work, a few suggestions are listed below to improve the current version.

1. Although I believe your work to be thorough and full, I advise making the discussion part stronger. Do the two RNP conformations that are demonstrated in vitro, for instance, actually exist in the natural state? If so, what physiological relevance does that have? If not, what external forces result in its formation?

We thank Reviewer #1 very much for the valuable suggestion. In the revised version of the manuscript, we have strengthened the Discussion part, which now at lines 177-197 contains a tentative interpretation of what could be the functional significance for the alternative assemblies in the filoviral RNP complex. Also, to this aim, new references have been added in support of the Discussion, including the works by Su et al. (2018, *Cell*), where a double-layered conformation of an RNP-like complex devoid of ssRNA was described for the filoviral species EBOV, and by Bharat et al. (2012, *PNAS*) and Beniac et al. (2012, *PLoS One*), where EBOV viral particles bearing multiple copies of the nucleocapsid were described.

2. In the course of your research, did you not notice the double-layered model at all during your research? Could it be that the proportion is too little to be taken into account during classification?

We agree that this is a very important question, and we are grateful to the Reviewer for raising this point, which encouraged us to undertake an in-depth assessment of the level of heterogeneity in our dataset through a reclassification of the 2D classes that did not show a clear polarity in their helical course during the first two rounds of 2D classification. Furthermore, also following the suggestion from the Reviewer #3, we have implemented our image processing workflow with a comparative analysis of the power spectra from the 2D classes, the 2D projections of the map from our dataset and those from the dataset of the published double-layered MARV RNP complex (EMD-31420). In addition, we have performed a comparative analysis of the inner and outer diameter profile in the two datasets. As shown by our results, which in the revised version of the manuscript appear in Supplementary Figures 1-5, no double-layered arrangement emerged from the averaging process, which agrees with our initial visual inspection of the micrographs. Admittedly, while we cannot rule out at this stage that such homogeneity in our dataset may be the result of the different sample preparation strategies adopted in our work with respect to the previous one by Fujita-Fujiharu and colleagues, we think that future structural studies *in situ* (on purified virions) and *in cellulo* (on infected or transfected cells) may be needed to elucidate the full diversity of

assemblies in filoviral RNP complexes.

3. What distinguishes your findings from those of the Ebola virus RNP, which likewise has a single-layer spiral structure?

We thank the Reviewer for raising this question. We kindly point out that, initially, we refrained from undertaking any comparative analysis of our RNP complex beyond the *Orthomarburgvirus* genus. Nonetheless, we recognize the importance of highlighting the similarities and differences of our findings with those reported for EBOV and other filoviral RNP complexes that are also in single-layer conformations. To this aim, in the revised version of the manuscript we have included in the Discussion part at lines 144-161 a new paragraph, where we briefly describe that - even though similar in their single-layer conformation - MARV, EBOV and LLOV RNP complexes do not share identical quaternary structures because of subtle differences in helical parameters and the presence (in EBOV and MARV) or the absence (in LLOV) of inter-strand non-axial interactions between NP protomers. In support of this part of the Discussion we have also added a panel (a) in Supplementary Figure 13, where the helical parameters of all previously reported structures of filovirus RNP complexes, including one from EBOV that appears in a double-layer conformation that is different from the one observed for MARV, are summarized. Furthermore, in support of our Discussion, we have included as new references the recent works by Fujita-Fujiharu et al. (2024) and by Watanabe et al. (2023), both in preprint at bioRxiv.org, where - in agreement with previous observation by Wan et al., (2017, *Nature*) - is reported the structural characterization of an outer layer to the RNP complex in the nucleocapsid, which is formed by subunits of VP24, VP35 and also by NP protomers not involved in ssRNA binding. We think that interaction of NP protomers of the RNP complex with the components of this nucleocapsid outer layer might have a role in modulating the helical parameters and the conformation of the RNP complex itself during the nucleocapsid biogenesis and maturation, as well as during its incorporation into nascent virions.

4. The addition of short ssRNA can cause NP to take on this helix structure, I wonder have you experimented with different nucleic acid lengths, and what were the outcomes?

Indeed, we did not perform such preliminary experiments for this project, just because we applied what we learnt from the experiments made during the obtainment of a structure in another project, the one of the cetacean morbillivirus RNP complex (Zinzula et al., 2021 *J Struct Biol*), where we observed that comparable results could be obtained by using ssRNA from 6 to 36 nt, by multiples of six nt, in length. While we fully recognize that such characterization would be interesting, especially in the context of the development of an *in vitro* biochemical assay to monitor the RNP complex assembly, in this project we opted for using directly an oligomer of 18 nt in length, because we reasoned that this could be, by length, the closest ssRNA to the tract that, at every step of the replication process, is putatively involved in interactions with the L-VP35 RNA-dependent RNA polymerase machinery: 6 nt in the L catalytic site, the previous 6 nt going to be wrapped again upon replacement into the NP_{core} groove, and the subsequent 6 nt going to be exposed upon displacement from the NP_{core}.

5. Please clarify whether it is possible to design tests to demonstrate the natural activity of RNP.

We thank the Reviewer for raising this point, which was not addressed in our discussion. To the best of our knowledge, we are not aware of any biochemical tool specifically developed to monitor the MARV RNP complex activity.

For EBOV, the biophysical characterization of different NP truncations with respect to their ability to form RNP-like oligomers was addressed by Su et al., (2018, *Cell*), who used dynamic light scattering (DLS) and hydrogen-deuterium exchange mass spectrometry (HDX-MS), and defined the regions that are important

to drive homo-oligomerization. Furthermore, they used a fluorescence polarization assay (FPA) to test binding of EBOV NP to the chaperone-like VP35 NPBP and to ssRNA. Similarly, Leung and colleagues (2015, *Cell Reports*) used radiolabeled nucleic acid transferred to nitrocellulose membrane via dot-blot assay to determine the affinity of EBOV NP for dsRNA and ssRNA, and ITC for binding of EBOV NP to VP35 NPBP. Also, Kirchdoerfer and colleagues (2015, *Cell Reports*) used ITC to measure binding of EBOV NP to VP35 NPBP. In addition, they used fluorescence anisotropy to measure the ssRNA binding capability between monomeric EBOV VP35-chaperoned NP and unchaperoned NP, testing either oligomerization competent (aa 1-450) or oligomerization incompetent (aa 34-367) NP truncations, and a FICT-labeled 18 nt-long ssRNA as substrate.

Regarding MARV, both Zhu and colleagues (2017, *Journal of Virology*) and Liu and colleagues (2017, *Journal of Virology*) used ITC to measure binding of NP to VP35 NPBP, and an electrophoretic mobility shift assay (EMSA) to visualize the binding of MARV NP to a biotin-labeled 15 nt-long ssRNA, either in the presence or in the absence of VP35 NPBP.

Although all highly valuable in providing insights on the mechanistic details of the NP-ssRNA and NP-VP35-NPBP interactions, none of the above-mentioned strategies was specifically designed to target the RNP complex assembly or its activity as scaffold for viral transcription and replication.

Nevertheless, it is our opinion that opportunities in this direction may come from adopting a methodological approach similar to the one reported by Lin and colleagues (2020, *Viruses*), who successfully coupled mini-genome system, VLP budding and bioluminescence resonance energy transfer (BRET)-based assay to monitor the formation of EBOV RNP complex *in vivo*.

Similarly, it is our opinion that drug candidates capable of disrupting or destabilizing the NP-NP and NP-RNA interactions that underly the MARV RNP complex formation – like the inhibitory ligands identified by Fu et al., (2016, *Scientific Reports*) – could be screened by taking advantage of bimolecular fluorescence complementation assays like the one that, for example, our group successfully validated to target EBOV and MARV VP35 homo-oligomerization (Zinzula et al., 2022, *iScience*). Therefore, in the revised version of the manuscript we have included all the above-mentioned relevant references and briefly addressed this point in the Discussion session, at lines 164-171.

Reviewer #3:

(Remarks to the Author):

The manuscript by Zinzula et al. presents a cryo-EM structure of the Marburg virus nucleocapsid core in a single-helical state. The determined structure is particularly interesting because it strongly contrasts with the double-helical assembly described last year by Fujita-Fujiharu et al. Furthermore, the presented helical arrangement is common for nucleocapsids of non-segmented negative strand RNA viruses and closely resembles in particular the one of another famous filovirus, EBOV. If the EMD-31420 map had not been published, the structure presented in the manuscript of Zinzula et al. might be sufficiently convincing. Admittedly, one can always suggest that the glaring differences between the EMD-31420 and the presented EMD-17838 maps lie in the method of sample preparation. However, even though Fujita-Fujiharu et al. write that «the double-helical structure reconstituted in this study may not faithfully recapitulate the structure in the MARV nucleocapsid core», given the current controversy surrounding the structure of the MARV nucleocapsid core, the authors of the present manuscript should provide more comprehensive evidence to support their conclusions.

- 1) Can it be that both single-helical and double-helical structures coexist in their preparation, as it was seemingly the case for Fujita-Fujiharu et al.?
For instance, considering that the data was acquired on a Titan Krios G4 with a Falcon 4, it would have been appropriate to collect more micrographs (say 20,000-30,000 instead of 5,518) in

order to perform a thorough 2D classification, beyond the one shown in Supplementary Figure 1. Indeed, one clearly sees from the Figure 1 that the preparation is extremely heterogeneous (which is quite common for the nucleocapsids of non-segmented negative strand RNA viruses), with very few helical filaments per micrograph suitable for high resolution image analysis. Importantly, in such cases it is crucial to specify how exactly the helical segments were selected, i.e. what was the box size and the percentage of overlap between consecutive segments. This information is missing in the manuscript, and in the figures sometimes a segment of one single turn is shown, sometimes 3, 4, 5 or 6. What was the segment size and overlap used for analysis, and were they changed during the processing or kept constant?

We thank the Reviewer for pointing out this fundamental aspect. Indeed, it was not possible for us to acquire more micrographs at the time of data collection, due to logistical constraints, and we thank the Reviewer for their understanding of such logistical impediment. Nevertheless, in spite of the intrinsic heterogeneity in terms of helix relaxation or compactness of the RNP complexes, during our analysis and visual inspection of the micrographs we observed neither co-existence between single- and double-layered conformations, nor major changes in helix diameters. However, in order to rule out such possibility and following the Reviewer's recommendation, we repeated a thorough 2D classification, which is now described in detail in the Methods session and also supported by new Supplementary Figures 1-3. In particular, particles with a box size of 256 pixels (px) and px size of 1.86 Å/px were extracted with helical priors and 16 asymmetrical units, which were used for the entire processing. In the revised version of the manuscript, we did our best to be consistent in the number of helical turns shown in the figures, in order to avoid confusion.

- 2) Following the first round of 2D classification shown in Supplementary Figure 1, one could select particles corresponding to classes showing helical side views and reclassify them iteratively, to see if all particles have the same diameter or if there is a heterogeneity at this level. To this end, it is essential to provide and analyse the histogram of diameter distribution (is it unimodal, centered on the inner-outer diameter of 174/299 Å of the presented EMD-17838 map, or bimodal, with a certain percentage of detected diameters corresponding to 255/330 Å as in EMD-31420, or are there also intermediate values present?).

We thank the Reviewer for this valuable suggestion, which we followed by performing a further 2D-classification step and an analysis of the particle-diameters, both now shown in Supplementary Figures 3 and 5, respectively. Specifically, after the last 2D-classification round, we selected from the remaining 23,887 particles those related to the class averages that did not show a clear polarity (therefore potentially ascribable to sample heterogeneity and co-existence of RNP complexes in different conformations). The resulting 3,956 particles were then subjected to another round of 2D-classification into 200 classes, at the end of which only 747 particles fell into class averages that retained unclear polarity, thereby assessing that heterogeneity in our dataset may account for about 3% of the final particles amount that was used for the subsequent 3D reconstruction steps. Nevertheless, no 2D-class averages showed clear double-layered conformation. These results are discussed in the main text at lines 59-62 and 70-74 of the revised version of the manuscript.

- 3) How do the sums of power spectra of particles in each class look like, which layer-lines do they show and how different or similar they are between classes? Which differences are expected between a single-helical and a double-helical structures?

Following the Reviewer's valuable recommendation, in the revised version of the manuscript we also performed analysis of the power spectra, now shown in Supplementary Figure 4. We provided a comparative overview of the class averages from the 2D-classification round for our structure, those from the real space projections of the map that we obtained, and those from the real space projections of the double-layered conformation of EMD-31420 dataset. As shown in Supplementary Figure 4, the power spectra of the 2D classes and projections of EMD-17833 (this work) display a similar pattern of layer lines, whereas the power spectra of the projection of EMD-31420 show a more complex pattern, whose layer lines markedly differ with respect to the previous ones, which agrees with the difference in conformation between the two samples that leads to a single-layered and double-layered arrangement of the RNP helix, respectively. The method and the results are discussed in the main text at lines 65-70 and 293-307, respectively, of the revised version of the manuscript.

- 4) *Idem*, the 3D classification and the generation of the initial model is insufficiently documented. The starting model was generated by 3D classification imposing 3 classes and a cylindrical reference, with the helical parameters of Fujita-Fujiharu et al. as an initial guess. The authors write in materials and methods that the class with the best resolution was selected as starting reference. Is this the class shown at the right side of the Supplementary Figure 1, under the 2D class averages? How many particles are present in this class? What do the other two classes look like?

We thank the Reviewer for raising these questions and soliciting us to reprocess our dataset and also to improve the Figure describing the single particle analysis workflow. We performed a reprocessing of the dataset throughout the workflow, now shown in a more comprehensive and detailed way in Supplementary Figure 1 and in the corresponding Methods section at lines 250-292 of the revised version of the manuscript. Furthermore, reprocessing our data led to higher resolution maps (3.1 Å instead of prior 3.4 Å), which we used to further improve our molecular model.

- 5) The authors then performed three rounds of 3D classification, 3D refinement, Bayesian polishing, and CTF-refinement, pooling the classes with the best resolution after each round, and resulting in a final dataset of 12,135 particles. Since no information on the shapes of the excluded 3D classes is provided, the presence of a low amount of particles corresponding to a double-helical structure cannot be excluded.

As for the above point, we thank the Reviewer for recommending a thorough revision of the description of our SPA workflow, which has been now edited both in the Methods section of the text (lines 250-292), as well as in the new version of Supplementary Figure 1 of the revised version of the manuscript, where information on the excluded 3D classes and their shapes is provided.

- 6) Since the structure solved correspond to NP_{core}-RNA only, please refrain from using the term nucleocapsid to avoid confusion with the full-length NP-RNA.

We thank very much the Reviewer for pointing out this important aspect, as we fully agree with the need to clearly distinguish between the terms “ribonucleoprotein complex” and “nucleocapsid”. Indeed, in agreement with the reviewer's point of view, we designate with the first term the macromolecular complex solely formed by the nucleoprotein and the viral genomic or antigenomic ssRNA, for the formation of which, moreover, the contribution provided by the sole NP_{core} domain alone is sufficient. Instead, with the second we refer to the macromolecular complex

formed by the ribonucleoprotein complex with the NP protein in its full length and, in the case of filoviruses, also the proteins VP35, VP24 and VP30, as they were previously recognized as structural components of the mature filoviral nucleocapsid.

In accordance with this distinction, we would like to kindly clarify that, throughout the manuscript, we never used the term "nucleocapsid" while referring to the structure that we determined and presented here, for which we instead strictly adhere to the term "ribonucleoprotein complex".

Specifically, on lines 15-16 and prior lines 125-126 (now lines 162-163), we state that our findings "expand the current knowledge about MARV genome packaging and nucleocapsid assembly" which, considering that the RNP formation is a necessary prerequisite for the nucleocapsid assembly, we think remains true. Likewise, on lines 30-31, we state that the "helical ribonucleoprotein (RNP) complex, [...] serves as scaffold for nucleocapsid formation", implying that the filoviral mature nucleocapsid will form on top of an assembled RNP complex. On prior line 124 (now line 150), the term "nucleocapsid" is referred to the corresponding cryo-ET map obtained by Wan et al. (2017, *Nature*, doi:10.1038/nature24490) from authentic MARV virions. Nevertheless, on prior line 79 (now line 94), in referring to the work by DiCarlo et al. (2007, *Virology Journal*, doi:10.1186/1743-422X-4-105), we wrote that the coiled-coil motif (of the NP core) was "reported as essential for viral RNA synthesis and nucleocapsid assembly". We agree that the term "nucleocapsid" in this statement could sound inappropriate and may bring confusion, and following the Reviewer's suggestion we have therefore substituted it with the term "RNP complex" in the revised version of the manuscript. We apologize for such inaccuracy and incoherence with respect to the terminology that we used elsewhere in the manuscript, which has been now made homogeneous and coherent in adherence to the above-mentioned definitions.

- 7) Line 48: «...ssRNA oligomers of six, or multiples of six, base pairs (bp) in length». You are using single strand RNA (ssRNA) with a length of six or multiple of six bases and not base pairs (there is no base pairing because it is a single strand RNA). Please replace «bp» by «bases» or «nucleotides» throughout the manuscript.

Correct. We thank very much the Reviewer for pointing out this inaccuracy in terminology and we apologize for the mistake. Accordingly, in the revised version of the manuscript we have substituted throughout the text the abbreviation "bp" (base pair) with the more appropriate "nt" (nucleotides).

- 8) It is inappropriate to say that «the in vitro reconstituted MARV RNP complex formed cylinders made up of NP-RNA stacked rings» while meaning a helical filament. Of note, some paramyxoviral and pneumoviral nucleocapsids can indeed form both helices and stacked rings, therefore employing inaccurate terminology may contribute to further confusion.

We are grateful for the Reviewer for highlighting this inaccuracy, agreeing that referring to "stacked rings" may lead to confusion. To avoid potential misinterpretations, in the revised version of the manuscript we have rephrased the sentence at line 55, substituting the term with "stacked spirals", which we consider more appropriate and in line with the Reviewer's suggestion.

- 9) At each helix turn, approximately 25.4 NP protomers follow one another with 14.1482° twist and 2.75 \AA rise, leading to a spiral with 69.85 \AA pitch». This sentence is unnecessary convoluted. In helices with one protomer per asymmetric unit, as the authors consider here because they don't take the RNA into account, there is no need to specify that the protomers follow one another. Instead, simply providing the values of twist and rise (or number of subunits per turn and pitch) is adequate.

We appreciate the Reviewer's advice to eliminate pleonasm and make the sentence more readable. Therefore, following the Reviewer's recommendation, in the revised version of the manuscript we opted for mentioning the number of subunits per turn and the approximate pitch in the main text (exact values are indicated in the Figure 1 and Extended Data Figure 4), modifying the sentence accordingly, which now appears at lines 76-77 as "Each helix turn is completed by approximately 25 NP subunits, and repeats with a pitch of about 70 Å".

- 10) The authors assembled NP_{core} on 18 base-long single strand RNA, with one NP_{core} bound to 6 consecutive bases. Therefore, in principle, if one considers the NP_{core}-RNA unit as an asymmetric unit of the helix, each asymmetric unit is composed of three NP_{core} protomers bound to one RNA segment. If the image analysis is performed with an asymmetric unit of 3 NP_{core} protomers, it would provide information about potential specific interactions with the RNA bases and the position of the base number 1, as previously done for paramyxoviruses (where an RNA of 6 bases was used for assembly and therefore the asymmetric unit was composed by only one N-RNA, thereby slightly simplifying the analysis). Hence, in order to fully exploit the potentials of their sample preparation strategy and gain novel information about the RNA binding beyond the one currently inferred from averaging over all protomers, it is highly advisable to revisit the analysis with three NP_{cores} per asymmetric unit. This would imply modifying the RNA model building and refinement accordingly.

We thank the Reviewer for the highly valuable suggestion. We have revisited the structural analysis considering three NP_{core} protomers per asymmetric unit, and consequently refined the map and built a new atomic model, which led to a higher final resolution (3.1 Å), and also to new values for the areas at the interface between NP_{core} protomers and between NP_{core} and ssRNA. However, the specific interactions were not resolved with the ssRNA showing longitudinal disorder – likely with register shifts of 6 and 12 nt – and the ssRNA was modelled accordingly. We have modified all figures where a monomeric or a tetrameric minimal unit of NP protomers was considered for the analysis, substituting them with trimeric and hexameric minimal units, respectively.

- 11) The authors claim that their reconstructed «...volume fits very well in that of the electron density map of a nucleocapsid molecular architecture determined by cryo-ET from authentic MARV virions (Supplementary Fig. 9)». First, to use the correct terminology, this is not an electron density map but a Coulomb potential map (or simply a cryo-ET map to avoid complicated terms). Second, this figure is not very convincing and the comparison between the maps deserves more explanation in the figure legend and the materials and methods section.

We thank the Reviewer for pointing out the incorrect terminology that we have used, for which we apologize. According to the Reviewer's recommendation, in the revised version of the manuscript, at line 149, we have re-phrased the sentence.

Regarding the prior Supplementary Figure 9 (now become Supplementary Figure 13), we would like to kindly point out that our intent was simply to qualitatively describe how the single-layered RNP complex can be accommodated within the nucleocapsid of an authentic MARV virion, of which the map used for comparison (EMD-3875) represents however only a portion of the outermost layer, whose protrusions have been attributed to protomers of the VP24 and VP35 proteins (Wan et al., 2017, *Nature*). Specifically, the detail that we wanted to highlight is how the groove that separates each rung from the subsequent one, well matches between the two maps. Admittedly, for a better appreciation of this match, would have been better to show as term of comparison the same superposition with the double-layered RNP complex. Initially, we refrained

from emphasizing too much the differences between our structure and the one reported by Fujita-Fujiharu and colleagues, so as not to dismiss the structural aspects of the work that preceded our own, which instead we appreciate and consider informative of a potential novel organization of the RNP complex. One that, albeit not clearly identified in the mature virion or the infected cells, we think that could still potentially have functional significance in the Marburg virus life cycle, as we speculate in the concluding sentences of the manuscript.

Nevertheless, fully appreciating the Reviewer's comment that the illustration was not convincing in its prior form, we have now modified it. Specifically, we added as a comparative term the same superposition with the double-layered RNP complex atomic model and map. Also, we defined planes that, passing through the same atom (OD2) of the side chain of Asp95 as the uppermost residue of the NP protomer, cut across each superposed map between two consecutive helical turns. As shown, in the case of the single-layered RNP complex, such planes separate subsequent helical turns and almost perfectly match the same groove in the two maps, whereas this is not appreciable at every helical turn in the case of the double-layered structure, because of the difference in the helix pitch. Furthermore, while in the case of the single-layer conformation the planes cut by half the density of the outermost protrusions of the nucleocapsid (recently ascribed to VP24, VP35 and an outer NP in the works by Fujita-Fujiharu et al., 2024, and Watanabe et al., 2023, both in preprint at bioRxiv), in the case of the double-layered conformation the same planes cut in between the protrusions that branch off two consecutive rungs. While such comparison, (given the low resolution of the nucleocapsid map and the poor cross correlation values from the fitting) cannot be conclusive about which of the two conformations is more compatible for incorporation in the mature virion, we think that it is informative to guide future analysis based on new high-resolution structural frameworks. Following the Reviewer's recommendation, we have discussed this part in the main text at lines 147-161, and also added sentences at lines 345-350 for better description of the analysis in the relevant Methods section, where we briefly describe how the overlapping of the two maps was performed. Finally, for further comparison and also to meet the suggestions from Reviewer #1, we have added to Supplementary Figure 13 the structures of RNP complexes from other filoviruses, such as Ebola and Lloviu viruses, which in recent publications were reported to be in single-layered conformation, and for Ebola virus also in a double-layered conformation different from the one observed for MARV (Su et al., 2018 *Cell*).

- 12) Finally, in the field of non-segmented negative strand RNA viruses, the N- and C-terminal domains of the nucleoprotein are called either N- and C-terminal domains, or N- and C-terminal lobes, or NTD and CTD. Please don't introduce new terms like foot and head, especially since from a structural point of view they cannot be applied consistently to the entire family.

We thank the Reviewer for the valuable recommendation. While we apologize for unintentionally contributing to confusion in the terminology used, we would like to kindly point out that no term was actually invented by us. Instead, the designations of "head lobe" and "foot lobe" to the N-terminal and C-terminal subdomains of the NP core domain of the filoviral nucleoprotein were introduced in the reference literature by the works of Leung et al. (2015, *Cell Reports*, doi:10.1016/j.celrep.2015.03.034) and Zhu et al. (2017, *Journal of Virology*, doi:10.1128/JVI.00996-17) when referring to the structures of the Ebola virus and Marburg virus NP respectively, and these terms have therefore been taken up by us in reference to these publications. Nonetheless, we welcome the Reviewer's suggestion to adhere to a more generally accepted terminology that meets the one used at the level of the entire Order *Mononegavirales*, and we have therefore modified accordingly the relevant passages of the text.